# Diet and Exercise Exert a Differential Effect on Glucose Metabolism Markers According to the Degree of NAFLD Severity

**DOI:** 10.3390/nu15102252

**Published:** 2023-05-10

**Authors:** Antonella Bianco, Isabella Franco, Ritanna Curci, Caterina Bonfiglio, Angelo Campanella, Antonella Mirizzi, Fabio Fucilli, Giuseppe Di Giovanni, Nicola Giampaolo, Pasqua Letizia Pesole, Alberto Ruben Osella

**Affiliations:** 1Laboratory of Epidemiology and Statistics, National Institute of Gastroenterology—IRCCS “S. de Bellis”, Via Turi, 70013 Castellana Grotte, BA, Italy; antonella.bianco@irccsdebellis.it (A.B.); isabella.franco@irccsdebellis.it (I.F.); ritanna.curci@irccsdebellis.it (R.C.); catia.bonfiglio@irccsdebellis.it (C.B.); angelo.campanella@irccsdebellis.it (A.C.); dottoressamirizzi@yahoo.com (A.M.); 2Department of Radiology, National Institute of Gastroenterology—IRCCS “S. de Bellis”, 70013 Castellana Grotte, BA, Italy; fabio.fucilli@irccsdebellis.it (F.F.); giuseppe.digiovanni@irccsdebellis.it (G.D.G.);; 3Laboratory of Clinical Pathology, National Institute of Gastroenterology—IRCCS “S. de Bellis”, 70013 Castellana Grotte, BA, Italy; letizia.pesole@irccsdebellis.it

**Keywords:** exercise, Mediterranean diet, non-alcoholic fatty liver disease, glycosylated hemoglobin, insulin resistance

## Abstract

Background: Non-Alcoholic Fatty Liver Disease (NAFLD) and Type 2 Diabetes (T2D) are highly prevalent diseases worldwide. Insulin Resistance (IR) is the common denominator of the two conditions even if the precise timing of onset is unknown. Lifestyle change remains the most effective treatment to manage NAFLD. This study aimed to estimate the effect of the Low Glycemic Index Mediterranean Diet (LGIMD) and exercise (aerobic and resistance) over a one-year period on the longitudinal trajectories of glucose metabolism regulatory pathways. Materials and Methods: In this observational study, 58 subjects (aged 18–65) with different degrees of NAFLD severity were enrolled by the National Institute of Gastroenterology—IRCCS “S. de Bellis”, to follow a 12-month program of combined exercise and diet. Results: The mean age was 55 ± 7 years old. Gender was equally distributed among NAFLD categories. There was a statistically significant main effect of time for glycosylated hemoglobin (Hb1Ac) over the whole period (−5.41, 95% CI: −7.51; −3.32). There was a steady, statistically significant decrease of HbA1c in participants with moderate and severe NAFLD whereas this effect was observed after the 9th month in those with mild NAFLD. Conclusions: The proposed program significantly improves glucose metabolism parameters, especially HbA1c.

## 1. Introduction

Nonalcoholic Fatty Liver Disease (NAFLD) is one of the most prevalent liver diseases, affecting up to 23% of the general population in Europe [1]. Obesity and insulin resistance are the main pathogenic factors linked to NAFLD, which is therefore a very frequent condition among type 2 diabetics (T2Ds) [2]. Furthermore, several studies have shown that liver dysfunction and T2D are closely related [3,4,5]. Although considerable efforts have been made to understand the immunopathogenic mechanism underlying NAFLD and its connections with T2D, data are still insufficient. However, insulin resistance (IR) seems to be one of the common denominators of both disorders [6,7]. The relationship between T2D, NAFLD, and IR remains intricate, making it difficult to identify which of these disorders is the cause and which the consequence. Since the liver is an essential regulator of glucose homeostasis, it is assumed that liver dysfunction may increase the risk of T2D [2]. On the other hand, it is possible that IR and T2D, by causing chronic inflammation and immunological alterations [8,9], may disturb liver function.

Our group has been studying the effects of lifestyle changes on NAFLD for some time. We observed the effects of the Low Glycemic Index Mediterranean Diet (LGIMD) alone [10], of aerobic exercise compared with resistance exercise [11], and of the combination of diet and exercise [12]. Specifically, we have shown that the association between LGIMD and aerobic exercise is among the most efficient treatments for subjects with NAFLD [12]. Furthermore, we have evidenced the synergistic effect of lifestyle interventions (diet and/or exercise programs) on the composition of the gut microbiota in patients with NAFLD [13].

In light of the evidence in the literature, we surmised that 12 months of supervised exercise and diet could improve NAFLD and the biomarkers related to glucose metabolism (especially HbA1c levels), helping to reduce the onset of T2D, as well as insulin resistance. The present study aimed to evaluate, during a one-year period, the effect of LGIMD and combined exercise (aerobic and endurance) on longitudinal trajectories of glucose metabolism regulatory pathways in subjects aged 18–65 years old affected by severe and moderate NAFLD as compared with those with mild or no NAFLD.

## 2. Materials and Methods

### 2.1. Participants

Details about the study have been published elsewhere [14]. Briefly, 58 subjects were recruited to the study conducted by the Laboratory of Epidemiology and Statistics of the National Institute of Gastroenterology—IRCCS “S. de Bellis”, Italy, to follow a 12-month program of combined exercise and diet. The project was started in March 2018 and ended in February 2020. Patients were enrolled in non-contemporary groups over the two years.

### 2.2. Study Design

In this observational study, a convenience sample was chosen, including subjects with NAFLD, referred by general practitioners, as candidates to follow a combined program of LGIMD and exercise.

The inclusion criteria were: (1) age between 18 and 65 years, (2) a confirmed diagnosis of NAFLD, (3) willingness to participate in three weekly classes in the gym for one year; and the exclusion criteria comprised: (1) absolute contraindications to exercise (acute cardiovascular disease, stroke, etc.) as outlined in the American College of Sports Medicine (ACSM) guidelines [15], (2) significant orthopedic or neuromuscular limitations, (3) unwillingness to participate or to follow an LGIMD for different reasons, (4) other coexisting liver diseases.

The study was conducted following the Helsinki Declaration and approved by the local Ethics Committee (DDG n.864; 17 December 2017).

### 2.3. Data Collection

During enrollment, participants signed informed consent statements and completed a structured questionnaire collecting data about sociodemographic aspects, medical history, and lifestyle. Physical activity information was collected using the validated International Physical Activity Questionnaire, Long Form (IPAQ-LF) [16]. The European Prospective Investigation into Cancer and Nutrition Food Frequency Questionnaire (EPIC FFQ) was used to probe alcohol intake and eating behavior [17]; in addition, two further questionnaires were administered to assess patient satisfaction with daily life: the SAT-P (Satisfaction Profile questionnaire) and the SF-36 questionnaire [18]. Blood samples and anthropometric measurements (weight, height, waist circumference) were collected by trained staff using standardized methods. More details are given in our previous study [14].

### 2.4. Measurements

NAFLD was assessed by liver ultrasound (LUS) (Esaote MyLab70 XVG device and the Convex 5-MHz probe) at baseline and then quarterly until the end of the project. A scoring system was adopted to obtain a semi-quantitative evaluation of the liver fat, based on three parameters: (1) contrast between liver and kidney parenchyma; (2) deep penetration of the ultrasound beam, and (3) sharpness of vascular structures, especially veins. Items were scored 0 to 2 [19,20,21].

NAFLD was then categorized as absent (0), mild (1–2), moderate (3–4), and severe (5–6). Liver size, margins, and echo-structure were assessed during the examination.

Laboratory measurements included triglycerides, total cholesterol, high-density lipoprotein cholesterol (HDL-C), low-density lipoprotein cholesterol (LDL-C), glucose, alanine transaminase (ALT), aspartate transaminase (AST), gamma glutamyl transferase (GGT), insulin, and glycosylated hemoglobin (Hb1Ac). As with LUS, biochemical assessments were performed at baseline and every trimester until month 12. In this work, only LUS and some biochemical markers were used in the analyses.

### 2.5. Dietary Plan

The Mediterranean diet is widespread in our area [22], so we attempted to reinforce the most important concepts of this dietary pattern in order to standardize the habits of the study subjects as much as possible, while allowing people to choose their own foods following the Mediterranean style.

Dietary recommendations were provided in the form of a booklet, with graphic explanations organized according to a traffic light system: with a list of foods that can be eaten often (green foods), sometimes (yellow foods), and rarely (red foods). There was no indication of food portions in the brochure.

Participants were given only general nutritional counseling and were advised to maintain their lifestyle. Patients were interviewed at regular intervals (3rd, 6th, and 9th month) by a trained nutritionist, who supervised their diet and provided counseling if necessary.

Diet composition is shown in Appendix A.

### 2.6. Exercise Protocol

#### 2.6.1. Fitness Assessment Tests

Three field tests were performed to assess the subjects’ basic conditions and establish the most appropriate training program. These included cardiorespiratory capacity, strength, and flexibility. Subjects repeated these tests every month until the end of the project. Cardiorespiratory fitness was assessed with the 2 km walk test [23], suitable for adults, whereas strength and flexibility fitness were assessed with the push-up test [24] (also called press-up test) and the sit and reach test [25], respectively.

#### 2.6.2. Aerobic Exercise and Resistance Training Combination

The combined training protocol (aerobic exercises combined with muscle strengthening) was performed three times a week for 12 months. Each session lasted approximately 1 h. We used Tanaka’s formula [26] to determine the intensity of the aerobic work and to calculate the one-repetition maximum (1-RM) for muscle work. All participants wore a heart rate monitor to constantly monitor the training intensity. The aerobic exercise included treadmill walking, rowing, and biking; the strengthening exercises included an initial free-body part with the aid of small pieces of equipment such as elastic bands, dumbbells, and sticks, followed by training on isotonic machines.

The training program began, for all subjects, with a conditioning period of 12 sessions in which low-intensity aerobic exercises (50–55% HRmax) and easy free-body strengthening exercises were performed. After this initial conditioning period, the training protocol was structured as shown in Table 1.

Each training phase lasted 12 sessions, at the end of which field tests were carried out to assess progress and move on to the next step.

The muscle-building part of the training was divided into three levels of difficulty (basic, intermediate, and advanced) [14]. The work intensity varied progressively from 65% to 75% of the 1RM.

To improve the monitoring of the participants’ capillary blood glucose (measured by Accu-Check^®^ digital glucometer) and blood pressure (by digital Omron^®^ sphygmomanometer) assessments were made during the training session (before and after). Oxygen saturation (as measured by digital finger Gima^®^ saturimeter) was also randomly assessed. Glycemic tests were performed following the American Diabetes Association (ADA) guidelines [27].

The presence of the participants at each training session was strictly registered and all participants underwent expert supervision for each exercise.

### 2.7. Statistical Analysis

Description and comparison between categories of NAFLD (absent/mild, moderate, and severe) and socio-demographic, lifestyle, and biological variables are reported as means (SD), performed by *t*-test, whereas categorical variables are reported as frequencies (percentages), and proportion differences were assessed using the χ^2^ test. All measures are in IU.

A Generalized Estimating Equation (GEE) [28] was performed to estimate the longitudinal trajectories of glucose (fast, and pre and post training), insulin, Homa-IR, and HbA1c. GEE models are particularly useful in biomedical studies to estimate mean changes in biomarker values while controlling for covariates, allowing correlations of response data (repeated measurements on each subject).

Model selection was made using diagnostic tools which allow the choice of the best performing set of covariates, the best working correlation structures, and all SE/robust options. As outcome variables were not normally distributed, a gamma distribution (link identity) for the response was assumed and an unstructured correlation matrix was set to the data. Age, sex and BMI (continuous variables) were included, as covariates. The results obtained are expressed on the original metric as mean ± 95% Confidence Interval (95% CI). We obtained outcome predictions using post-estimation tools; this analysis is graphically displayed.

Stata statistical software v. 17.0 (StataCorp, 4905 Lakeway Drive, College Station, TX 77845, USA) was used to perform statistical analyses. The official command xtgee and the user-written contribution QIC (criterion for model selection in GEE analyses) were used.

## 3. Results

In total, fifty-eight subjects, 41% with a moderate grade of NAFLD, were included in the study. Participants’ characteristics are shown in Table 2.

Mean age was 55 ± 7 years old. Men and women were equally distributed among NAFLD categories. All subjects had a high BMI and high total cholesterol levels. Altered values of triglycerides, ALT, GGT, AST, glucose, HbA1c and Homa-IR were observed especially in subjects with severe NAFLD.

Most subjects with a moderate and severe grade of NAFLD had impaired glucose tolerance (71%). A small proportion of the participants were smokers.

Table 3 showed the changes in hematochemical values over time among subjects with different grades of NAFLD.

Importantly, some subjects experienced a remarkable change in their condition, from more severe forms of NAFLD to the absence of NAFLD, as from the third month.

BMI values tended to decrease up to the sixth month in all NAFLD categories, then rose again, especially in subjects with moderate NAFLD. This trend was not observed among subjects with severe NAFLD, who all had a lower BMI at the end of the project than at the beginning.

In subjects with mild and moderate NAFLD, glucose values fell but tended to rise again at the ninth and twelfth months, respectively, although they never reached the initial values. In subjects with severe NAFLD, these values decreased steadily to about 24% of the initial value. At the end of the project, blood glucose levels had reached values considered normal in all groups.

Homa-IR values decreased, especially in subjects with mild and moderate NAFLD. In the first group of participants, there was an abnormal rise between the sixth and ninth months, then a reduction by the end of the year; in this NAFLD category, Homa-IR values were normal at the twelfth month. Subjects with moderate NAFLD exhibited a decreasing trend of Homa-IR values (but a rise in the sixth month). By the end of the study, there was a 33% reduction in Homa-IR values.

In subjects with severe NAFLD, Homa-IR values showed not only a smaller decrease than in the other categories but also an irregular trend.

Insulin values fluctuated inconsistently in all groups, whereas Hb1Ac, AST and ALT showed a decreasing trend in all NAFLD categories. Table 4 shows the results of GEE modeling.

There was a statistically significant main effect of time for glucose as from the ninth month (−8.56, 95% CI: −14.34; −2.77, −8.54, 95% CI: −15.15; −1.94), for insulin at the third month (−2.39, 95% CI: −4.50; −0.28) and for Hb1Ac throughout the whole period (−1.93, 95% CI: −3.59; −0.27, −3.84, 95%CI: −5.55; −2.13, −4.80. 95% CI: −6.68; −2.91, −5.41, 95% CI: −7.51; −3.32).

Table 5 shows the contrast among mean expected values for the modification effect between time and NAFLD of different severity grades, at different time points as compared with the initial values.

The modification effect between time and grade of NAFLD severity for glucose did not reach statistically significant values except in the severe NAFLD group at the twelfth month (−16.74 95%CI −32.33; −1.15). The same trend was observed for insulin, except in the moderate NAFLD group at the third (−4.09 95%CI −7.42; −0.76) and twelfth months (−5.33 95%CI 9.59;−1.07) and for Homa-IR in the moderate NAFLD group at the twelfth month (−1.58 95%CI −2.90;−0.26).

Hb1Ac showed a decreasing trend of the modification effect of time and grade of NAFLD severity mainly as from the third month, for subjects with moderate and severe NAFLD. This effect was not observed in subjects with mild NAFLD.

Longitudinal trajectories of fasting blood glucose levels in all groups are shown in Figure 1.

Overall, there was a reduction in glucose levels until the ninth month. This trend was not evident in the severe NAFLD category.

Longitudinal trajectories of insulin levels in all groups are also shown in Figure 1. For subjects with absent and mild NAFLD, the insulin (at different levels) trend was similar. Subjects with moderate NAFLD exhibited stable levels, and with severe NAFLD, a rise was evident as from the ninth month.

Longitudinal trajectories of Homa-IR in all groups are shown in Figure 2. 

Homa-IR seems to reflect the insulin trajectories except in subjects with mild NAFLD, who showed a decrease in the first part of the period and a rise at the end.

Longitudinal trajectories of Hb1Ac in all groups are shown in Figure 2. Overall, there was a decreasing trend for all NAFLD categories until at least the ninth month.

Table 6 shows the expected trend in pre- and post-exercise blood glucose values by time in the entire sample.

Blood glucose levels measured before training decreased statistically significantly from the fifth month until the end, reaching the maximum reduction at the eleventh month (−10.28, 95% CI: −13.06; −7.50). Post-workout values also decreased, to a minimum in the eleventh month (−5.67, 95% CI: −7.69, −3.65), but already showed a significant reduction by the second month of activity.

## 4. Discussion

NAFLD in the early stages is often described as a “benign condition” in the context of liver disease. However, the effects of steatosis extend beyond the liver [29]. Since the liver plays a crucial role in regulating glucose homeostasis, studies have shown that individuals with diagnosed NAFLD have a two-fold higher risk of developing T2D [3] and insulin resistance, and a greater risk of developing cancer [30] and cardiovascular [31,32] and kidney disease [33], in particular when associated with T2D [34]. Furthermore, steatosis appears to be an early predictor of metabolic disorders [35,36]. Our data confirmed this association, as 71% of the subjects with severe and moderate NAFLD included in our project had impaired glucose tolerance at baseline, as measured by the Homa index. This study has shown that, in subjects with different degrees of NAFLD, LGIMD combined with an exercise program (aerobic and resistance training) does not only improve NAFLD status and its related biomarkers but also has a beneficial effect on glucose metabolism regulatory pathways.

A multidisciplinary lifestyle intervention, including diet therapy and exercise, is the cornerstone of NAFLD management [37]. It has been shown that subjects who followed a Mediterranean diet showed a better reduction in liver transaminases and body mass index (BMI), and improved IR [38,39]. In fact, the intake of monosaturated FA present in foods such as olive oil has been shown to improve insulin sensitivity and liver fat in prediabetic patients [40] and patients with NAFLD [41]. The Mediterranean diet, featuring rich contents of antioxidants, unsaturated fatty acids, and fiber, and low quantities of animal protein and saturated fat, can improve or prevent Metabolic Syndrome, T2D, or cardiovascular diseases [42], and can also improve the lipid profile and IR, thus preventing NAFLD-related diseases [43]. In this sense, our previous study showed that the LGIMD diet, which satisfies these requirements, is effective in patients with NAFLD [10]. As the Mediterranean diet is prevalent in our area, we believed it would be easier for participants to adhere to the diet plan because it was similar to the food they usually ate, thus ensuring a better compliance.

Studies conducted with exercise intervention alone, with or without weight loss, have also been shown to improve the clinical parameters of NAFLD [44]; however, this effect is stronger when subjects lose weight [45].

Furthermore, current evidence suggests that both the Mediterranean diet and physical activity are able to modify the gut microbiota [46] which, in turn, improves glucose metabolism [47], then the host health, even in individuals with NAFLD [13].

Our study showed that remarkably, some subjects lost their NAFLD status after three months of LGIMD combined with an exercise program; ALT and AST values also decreased continuously, confirming the improved NAFLD status. In addition, between the third and the sixth months, there was a reduction in BMI in all subjects, regardless of the degree of NAFLD. Several studies have shown a positive association between changes in BMI and changes in liver fat content [45,46,47,48], showing that for every 1% decrease in body weight, there is a 1% decrease in liver fat content. Only in subjects with severe NAFLD did the body weight, as measured by BMI, remain lower than at baseline. Fasting blood glucose levels were also reduced in all subjects, but only in participants with severe NAFLD did they reach a statistically significant reduction, which could be due to the higher baseline level, as also occurred for the BMI. In fact, it is known that the higher the initial BMI, the greater the reduction [49].

Additionally, we observed a significant decrease in blood glucose levels immediately after combined training in all subjects, but this was more pronounced and statistically significant in participants with severe NAFLD, and particularly in subjects with moderate NAFLD. The ACSM [50] showed that endurance exercise reduces blood glucose levels for at least 24 h after exercise in subjects with low fasting blood glucose. In our study, during the last month of training, these values tended to rise, but in subjects with moderate NAFLD, these values continued to fall until month 12. This could be due to the reduction in the number of subjects with mild and severe NAFLD participating in the project. Pre-exercise glycemia also decreased over time, especially in subjects with moderate NAFLD, highlighting the long-term benefits of exercise, especially when supervised and controlled by experts. A study by Li et al. [51] showed that lifestyle intervention could lead to some metabolic memory and the effects on insulin secretion or sensitivity could be maintained well beyond the duration of the physical activity and diet intervention.

Insulin values in subjects with mild and moderate NAFLD were also reduced; consequently, Homa-IR index values declined. It is known that exercise, through muscle contraction, improves glucose uptake through the GLUT-4 receptor by increasing muscle insulin sensitivity [52]. Furthermore, exercise increases muscle glucose storage in the form of glycogen [53,54]. In subjects with severe NAFLD, however, insulin and Homa-IR index values tended to increase in the latter part of the project. Liu et al. [55] pointed out in their meta-analysis that a reduction in insulin levels was only noted in diabetic subjects who had followed a high-intensity resistance exercise program. In fact, it is known that modifying the FITT (Frequency, Intensity, Time, and Type) parameters of exercise can optimize the effect of reducing glucose metabolism markers in different populations [56,57]. Therefore, it is likely that in subjects with severe NAFLD (most of whom had insulin resistance), the exercise we proposed was not sufficiently intense to produce positive changes in insulin levels. The abnormal increase in insulin values and the Homa index observed in subjects with mild or moderate NAFLD around the ninth month was probably due to a reduction in the number of participants (drop-out) and a decreased compliance to the program (diet and exercise). The difficulty of maintaining healthier lifestyle changes over long periods is well known [58,59].

In this study, a statistically significant reduction in HbA1c values was observed in all subjects, especially in participants with severe and moderate NAFLD. Currently, much attention has been paid to the role of HbA1c in identifying impairments in glucose metabolism in individuals without diabetes [60]. Furthermore, a recent meta-analysis has shown that an optimal HbA1c level (5.0–6.0%) is important in preventing the risk of all-cause mortality in non-diabetic populations [61]. Since microvascular complications of diabetes are present in the early stages of the disease, monitoring HbA1c levels should also be extended to non-diabetics [62].

As our exercise program appears to be efficient in reducing HbA1c levels, the beneficial effect could be due to the increase in the muscle capillary network and blood flow, which promotes glucose synthesis and increases the clearance of free fatty acids [63]. With fewer available blood glucose molecules, the binding of glucose to hemoglobin heteroprotein decreases, resulting in a reduction in HbA1c [62]. Some studies have suggested that HbA1c levels may reach a plateau between 8 and 12 weeks [64], confirming that <12-week physical activity interventions are more effective in reducing HbA1c among non-diabetics, as stated by Cavero-Redondo et al. [62]. On the other hand, Wrench et al. [65] observed a significant reduction in HbA1c values after >12 weeks of aerobic exercise in diabetic subjects. Our results are in line with those shown by Wrench et al. [64], probably because 71% of our subjects had insulin resistance.

The ADA recommends the measurement of HbA1c as the standard for monitoring diabetes, as it is directly correlated with average plasma glucose concentrations [66]. Numerous studies showing a decrease in blood glucose concentrations support the evidence provided by the HbA1c biomarker that exercise programs have a beneficial effect on glycemic control in diabetic [65] and non-diabetic subjects [62].

Our results consolidate the evidence supporting exercise as a powerful strategy to prevent type 2 diabetes, especially in industrialized countries, and in individuals with NAFLD, to which diabetes is closely related. The most important strength of this study is the supervised nature of the program, controlled by personnel specialized in diet and in training. Moreover, due to the reduced number of participants in each group, the presence of two trainers in each session guaranteed a personalized relationship with each participant. Furthermore, a programmed interview with the nutritionist reinforced the Mediterranean dietary pattern in this area for each participant. In addition, the exercise protocol was constructed following specific FITT parameters, in accordance with the guidelines of the major international associations [67].

Limitations include the missing sample size calculation and, as a consequence, the small sample size. However, similar study designs showed that a comparable sample size was sufficient to find significant differences in the nutritional [10] and physical activity exposures [11] conducted to lower the degree of severity of NAFLD. Several variables were used to adjust estimates but additional adjustments may be necessary. However, rigorous statistical procedures were performed to choose the best set of adjusting variables, as well as the best working correlation structures.

## 5. Conclusions

Following an LGIMD and a combined exercise program (aerobic + resistance exercise) of a weekly volume of at least 180 min, at moderate intensity, for 12 months, leads to significant improvements in glucose metabolism parameters in different ways, depending on the degree of NAFLD. Glucose values are significantly reduced in subjects with severe NAFLD, and insulin and Homa values are reduced in subjects with moderate NAFLD. In contrast, all subjects, regardless of the degree of severity, benefit from an improvement in NAFLD status and related markers, especially HbA1c values. The latter showed a greater drop in subjects with severe NAFLD than in the other groups. We believe that further studies are necessary to test whether differentiating the FITT parameters according to the degree of NAFLD of the participants could lead to greater improvements in each individual of glucose metabolism biomarkers. Monitoring HbA1c levels could also be useful in non-diabetic populations to check for the onset of T2DM in at-risk individuals, prompting early intervention.

## Figures and Tables

**Figure 1 nutrients-15-02252-f001:**
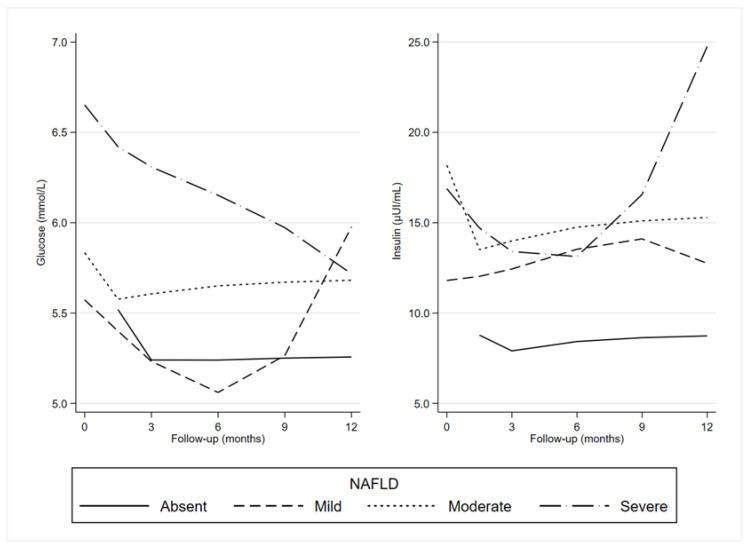
Generalized Estimating Equation (GEE): predictive margins of glucose and insulin by the degree of NAFLD severity and time. NAFLD: Non-Alcoholic Fatty Liver Disease.

**Figure 2 nutrients-15-02252-f002:**
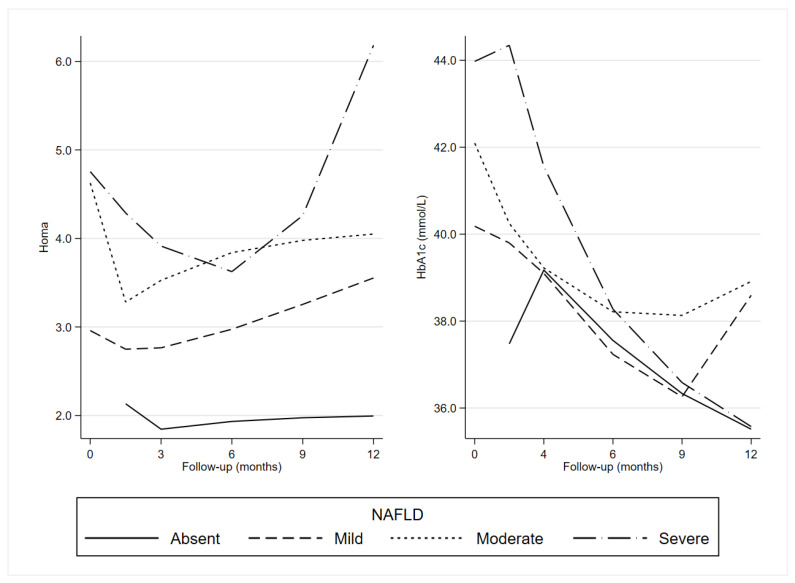
Generalized Estimating Equation (GEE): predictive margins of Homa-IR and HbA1c by the degree of NAFLD severity and time. Homa: Homeostatic Model Assessment for Insulin Resistance; HbA1c: Glycosylated Haemoglobin; NAFLD: Non-Alcoholic Fatty Liver Disease.

**Table 1 nutrients-15-02252-t001:** Exercise program with progressive changes in duration and intensity.

Period	Intensity
Conditioning	Intensity at 50–55% HRmax ^a^ + exercises at basic level
1st Step	5% increase in intensity (55/60% HRmax) + introduction of intermediate level exercises
2nd Step	Increased duration of aerobic work by 5′ (tot.15′) + reduced conditioning
3rd Step	5% increase in intensity (60/65% HRmax) and maintenance of intermediate level exercises
4th Step	Increased duration by 5′ of the strengthening work (tot.15′) + reduction in cool down time
5th Step	5% increase in intensity (65/70% HRmax) + intermediate and advanced level exercises
6th Step	Increased duration by 5′ of aerobic work (tot.20′)
7th Step	5% increase in intensity (70/75% HRmax) + advanced level exercises
8th Step	Increased duration by 5′ of the strengthening work (tot.20′) + reduction in all the other phases (aerobic excluded)

^a^ HRmax: Maximum Heart Rate.

**Table 2 nutrients-15-02252-t002:** Characteristics of participants at baseline.

	Whole Sample		NAFLD		
		Mild	Moderate	Severe	*p* Value ^§^
N	58	23	24	11	
Age (years) *	55.16 (7.36)	55.75 (7.07)	56.33 (6.58)	51.33 (8.90)	0.16
Sex **					
Female	29 (50%)	12 (52%)	12 (50%)	5 (45%)	0.94
Male	29 (50%)	11 (48%)	12 (50%)	6 (55%)	
SBP (mmHg) *	128.77 (12.22)	126.52 (10.27)	131.74 (14.59)	127.27 (10.09)	0.32
DBP (mmHg) *	81.58 (9.12)	81.09 (7.68)	82.39 (11.27)	80.91 (7.35)	0.86
BMI (kg/m^2^) *	31.84 (5.00)	30.20 (4.28)	32.37 (5.17)	34.11 (5.27)	0.078
Triglycerides (mmol/L) *	1.74 (1.01)	1.69 (1.01)	1.66 (1.08)	2.04 (0.86)	0.55
Total Cholesterol (mmol/L) *	5.32 (0.98)	5.49 (0.85)	5.13 (0.94)	5.40 (1.31)	0.45
HDL-C (mmol/L) *	1.16 (0.29)	1.19 (0.32)	1.15 (0.28)	1.12 (0.27)	0.79
LDL-C (mmol/L) *	3.36 (0.83)	3.52 (0.60)	3.22 (0.78)	3.34 (1.27)	0.46
ALT (μkat/L) *	0.52 (0.29)	0.36 (0.10)	0.58 (0.31)	0.75 (0.36)	<0.001
GGT (U/L) *	0.45 (0.42)	0.43 (0.47)	0.39 (0.31)	0.61 (0.49)	0.36
AST (μkat/L) *	0.43 (0.16)	0.35 (0.08)	0.46 (0.17)	0.53 (0.21)	0.003
Glucose (mmol/L) *	5.88 (1.52)	5.41 (0.83)	5.84 (1.46)	6.97 (2.21)	0.017
Hb1Ac (mmol/L) *	41.50 (10.30)	38.52 (5.43)	42.50 (11.53)	45.55 (13.91)	0.15
Homa-IR *	3.94 (2.29)	2.73 (1.16)	4.60 (2.85)	5.06 (1.58)	0.003
Insulin * (µUI/mL)	15.33 (9.25)	11.30 (4.55)	18.08 (11.84)	17.73 (7.60)	0.024
Smoker **					
Never/Former	52 (90%)	21 (91%)	21 (88%)	10 (91%)	0.90
Current	6 (10%)	2 (9%)	3 (13%)	1 (9%)	
Resistent Insulin **					
Homa < 2.5	17 (29%)	13 (57%)	3 (13%)	1 (9%)	0.001
Homa ≥ 2.5	41 (71%)	10 (43%)	21 (88%)	10 (91%)	
Diabetes **					
No	41 (71%)	20 (87%)	15 (63%)	6 (55%)	0.078
Yes	17 (29%)	3 (13%)	9 (38%)	5 (45%)	

NAFLD: Non-Alcoholic Fatty Liver Disease; SBP: Systolic Blood Pressure; DBP: Diastolic Blood Pressure; BMI: Body Max Index; HDL-C: High-Density Lipoprotein; LDL-C: Low-Density Lipoprotein; ALT: Alanine Amino Transferase; GGT: Gamma Glutamyl Transferase; AST: Aspartate Transaminase; HbA1c: Glycosylated Hemoglobin; Homa-IR: Homeostatic Model Assessment for Insulin Resistance. * Mean(SD), ** Nr(%), ^§^
*p*-values refer to ANOVA or chi-square test between categories of NAFLD.

**Table 3 nutrients-15-02252-t003:** Biomarker values observed at different follow-ups and different degrees of NAFLD severity.

	NAFLD				
	Absent	Mild #	Moderate	Severe	*p*-Value *^§^*
	Mean (SD)	Mean (SD)	Mean (SD)	Mean (SD)	
BMI					
Baseline		30.20 (4.28)	32.37 (5.17)	34.11 (5.27)	0.078
3 months	28.44 (4.42)	30.22 (3.23)	31.77 (5.35)	31.49 (3.47)	0.31
6 months	29.51 (7.92)	29.12 (3.83)	31.97 (4.11)	32.64 (2.29)	0.42
9 months	27.09 (3.91)	30.16 (4.54)	32.17 (5.68)	35.12 (5.34)	0.040
12 months	29.17 (4.13)	31.98	34.54 (6.01)	32.01 (4.13)	0.34
SBP (mmHg)					
Baseline		126.52 (10.27)	131.74 (14.59)	127.27 (10.09)	0.32
3 months	120.00 (11.55)	127.50 (9.57)	121.43 (12.39)	120.00 (12.65)	0.75
6 months	131.00 (15.95)	121.43 (13.45)	125.56 (14.23)	126.00 (11.40)	0.59
9 months	128.00 (10.33)	127.50 (5.00)	125.38 (8.77)	122.50 (12.58)	0.77
12 months	118.00 (10.95)	130.00	129.55 (10.11)	132.50 (9.57)	0.16
DBP (mmHg)					
Baseline		81.09 (7.68)	82.39 (11.27)	80.91 (7.35)	0.86
3 months	77.50 (5.40)	73.75 (4.79)	77.50 (7.64)	75.00 (5.48)	0.66
6 months	81.00 (9.66)	78.57 (9.00)	78.89 (5.57)	84.00 (5.48)	0.52
9 months	78.00 (6.32)	76.25 (7.50)	78.85 (5.83)	83.75 (7.50)	0.38
12 months	77.00 (10.95)	80.00	81.36 (7.10)	82.50 (5.00)	0.71
Glucose (mmol/L)					
Baseline		5.41 (0.83)	5.84 (1.46)	6.97 (2.21)	0.017
3 months	5.19 (0.69)	4.93 (0.37)	5.65 (1.05)	6.86 (1.28)	0.008
6 months	5.42 (1.05)	5.11 (0.81)	5.74 (1.22)	6.34 (1.53)	0.30
9 months	5.03 (0.40)	5.30 (0.82)	5.55 (0.78)	5.30 (0.38)	0.33
12 months	5.07 (0.63)	5.16	5.70 (0.72)	5.32 (0.22)	0.32
Homa-IR					
Baseline		2.73 (1.16)	4.60 (2.85)	5.06 (1.58)	0.003
3 months	1.88 (1.55)	1.83 (0.37)	3.83 (4.41)	5.12 (2.26)	0.25
6 months	1.78 (1.05)	2.53 (1.46)	5.77 (7.80)	3.56 (1.10)	0.27
9 months	1.37 (0.50)	5.09 (2.25)	3.01 (1.34)	4.94 (2.73)	<0.001
12 months	1.47 (0.89)	1.64	3.08 (1.26)	6.79 (3.19)	0.002
Insulin (µUI/mL)					
Baseline		11.30 (4.55)	18.08 (11.84)	17.73 (7.60)	0.024
3 months	7.68 (5.37)	8.41 (1.81)	14.40 (12.77)	17.23 (8.46)	0.22
6 months	7.13 (3.61)	10.59 (4.65)	19.67 (19.60)	12.74 (3.75)	0.13
9 months	6.06 (2.05)	22.90 (13.79)	12.20 (5.25)	21.22 (12.62)	<0.001
12 months	6.25 (3.12)	7.16	12.21 (4.89)	29.10 (14.46)	0.001
Hb1Ac (mmol/L)					
Baseline		38.52 (5.43)	42.50 (11.53)	45.55 (13.91)	0.15
3 months	38.80 (7.36)	37.75 (7.41)	38.46 (6.71)	52.17 (13.29)	0.003
6 months	37.50 (4.06)	37.14 (5.01)	38.39 (10.26)	40.00 (5.29)	0.92
9 months	35.50 (5.06)	41.00 (12.08)	40.54 (7.00)	32.50 (3.70)	0.12
12 months	35.60 (3.36)	29.00	40.45 (8.62)	32.75 (5.44)	0.19
ALT (μkat/L)					
Baseline		0.35 (0.08)	0.46 (0.17)	0.53 (0.21)	0.003
3 months	0.36 (0.08)	0.33 (0.03)	0.40 (0.11)	0.46 (0.08)	0.14
6 months	0.39 (0.10)	0.40 (0.11)	0.40 (0.10)	0.40 (0.05)	0.99
9 months	0.33 (0.08)	0.34 (0.04)	0.38 (0.10)	0.50 (0.19)	0.066
12 months	0.35 (0.09)	0.32	0.33 (0.08)	0.45 (0.12)	0.17
AST (μkat/L)					
Baseline		0.36 (0.10)	0.58 (0.31)	0.75 (0.36)	<0.001
3 months	0.36 (0.14)	0.36 (0.14)	0.42 (0.16)	0.59 (0.15)	0.035
6 months	0.37 (0.11)	0.41 (0.14)	0.45 (0.23)	0.53 (0.14)	0.39
9 months	0.27 (0.05)	0.40 (0.17)	0.42 (0.24)	0.72 (0.40)	0.015
12 months	0.33 (0.09)	0.25	0.36 (0.11)	0.61 (0.38)	0.10
GGT (U/L)					
Baseline		0.43 (0.47)	0.39 (0.31)	0.61 (0.49)	0.36
3 months	0.28 (0.24)	0.37 (0.23)	0.37 (0.22)	0.33 (0.09)	0.76
6 months	0.21 (0.08)	0.37 (0.29)	0.38 (0.21)	0.41 (0.12)	0.15
9 months	0.27 (0.15)	0.33 (0.14)	0.38 (0.28)	0.69 (0.55)	0.10
12 months	0.25 (0.05)	0.40	0.31 (0.17)	0.71 (0.51)	0.058
Triglycerides (mmol/L)					
Baseline		1.69 (1.01)	1.66 (1.08)	2.04 (0.86)	0.55
3 months	1.10 (0.49)	1.85 (0.67)	1.69 (1.19)	1.09 (0.64)	0.26
6 months	1.35 (0.97)	1.40 (0.67)	1.64 (0.57)	1.51 (0.71)	0.74
9 months	1.14 (0.71)	1.00 (0.19)	1.88 (1.03)	1.77 (0.81)	0.12
12 months	1.24 (0.19)	1.38	1.61 (0.98)	3.53 (1.73)	0.021
Total Cholesterol (mmol/L)					
Baseline		5.49 (0.85)	5.13 (0.94)	5.40 (1.31)	0.45
3 months	5.10 (0.91)	5.13 (0.90)	5.08 (0.79)	3.69 (0.74)	0.004
6 months	5.30 (0.81)	4.71 (1.08)	5.00 (1.03)	5.53 (1.36)	0.50
9 months	5.18 (0.89)	3.96 (2.08)	5.47 (0.87)	4.44 (0.64)	0.076
12 months	5.62 (0.82)	4.73	5.29 (1.32)	5.47 (0.97)	0.89
HDL					
Baseline		1.19 (0.32)	1.15 (0.28)	1.12 (0.27)	0.79
3 months	1.24 (0.27)	0.98 (0.24)	1.12 (0.32)	1.16 (0.19)	0.48
6 months	1.21 (0.16)	1.07 (0.39)	1.11 (0.21)	1.23 (0.39)	0.58
9 months	1.29 (0.32)	0.98 (0.16)	1.20 (0.33)	1.01 (0.26)	0.27
12 months	1.36 (0.29)	0.93	1.23 (0.38)	0.92 (0.20)	0.24
LDL					
Baseline		3.52 (0.60)	3.22 (0.78)	3.34 (1.27)	0.46
3 months	3.36 (0.66)	3.30 (0.81)	3.19 (0.85)	2.03 (0.43)	0.008
6 months	3.46 (0.63)	3.00 (0.93)	3.14 (0.88)	3.62 (1.32)	0.53
9 months	3.37 (0.59)	2.53 (2.03)	3.40 (0.61)	2.62 (0.62)	0.19
12 months	3.70 (0.51)	3.18	3.40 (0.84)	2.94 (1.31)	0.64

NAFLD: Non-Alcoholic Fatty Liver Disease; BMI: Body Max Index; SBP: Systolic Blood Pressure; DBP: Diastolic Blood Pressure; Homa-IR: Homeostatic Model Assessment for Insulin Resistance; HbA1c: Glycosylated Haemoglobin; ALT: Alanine Amino Transferase; AST: Aspartate Transaminase; GGT: Gamma Glutamyl Transferase; HDL-C: High-Density Lipoprotein; LDL-C: Low-Density Lipoprotein. # Only 1 subject at time 12 months with mild steatosis. ^§^
*p*-value refers to ANOVA test between categories of NAFLD.

**Table 4 nutrients-15-02252-t004:** Generalized Estimating Equation (GEE): expected values for different markers by time and degree of NAFLD severity.

	Glucose		Insulin		Homa-IR		Hb1Ac	
	β	95% CI	β	95% CI	β	95% CI	β	95% CI
Follow-up								
Baseline	0.00		0.00		0.00		0.00	
3 month	−4.65	−9.74; 0.44	−2.39 *	−4.50; −0.28	−0.59	−1.28; 0.09	−1.93 *	−3.59; −0.27
6 month	−3.74	−9.05; 1.57	−1.72	−3.88; 0.44	−0.39	−1.09; 0.31	−3.84 **	−5.55; −2.13
9 month	−8.56 **	−14.34; −2.77	−0.91	−3.38; 1.57	−0.39	−1.16; 0.39	−4.80 **	−6.68; −2.91
12 month	−8.54 **	−15.15; −1.94	−1.82	−4.62; 0.99	−0.72	−1.57; 0.13	−5.41 **	−7.51; −3.32
NAFLD								
Absent	0.00		0.00		0.00		0.00	
Mild	−3.47	−9.73; 2.78	0.96	−1.26; 3.17	0.23	−0.44; 0.89	−1.72	−3.80; 0.36
Moderate	−0.11	−6.32; 6.09	2.39	−0.02; 4.80	0.92 **	0.19; 1.66	−1.14	−3.23; 0.95
Severe	10.29 *	1.89; 18.68	2.28	−1.13; 5.69	1.35 *	0.22; 2.47	0.32	−2.46; 3.09

Adjusted for age, sex and BMI ** *p* < 0.001 * *p* < 0.05. Homa-IR: Homeostatic Model Assessment for Insulin Resistance; HbA1c: Glycosylated Haemoglobin; NAFLD: Non-Alcoholic Fatty Liver Disease.

**Table 5 nutrients-15-02252-t005:** Generalized Estimating Equation (GEE): contrasts of the combination of degree of NAFLD severity and time from baseline.

	Glucose		Insulin		Homa-IR		HbA1c	
Contrast	β	95% CI	β	95% CI	β	95% CI	β	95% CI
(3 ms vs. t0) Mild	−7.75	−20.46; 4.96	−1.49	−5.58; 2.60	−0.45	−1.64; 0.74	0.91	−3.50; 5.32
(6 ms vs. t0) Mild	−8.85	−18.66; 0.96	−0.23	−3.83; 3.36	−0.23	−1.25; 0.79	−2.87	−6.02; 0.29
(9 ms vs. t0) Mild	−7.17	−19.98; 5.63	3.73	−2.77; 10.23	0.74	−1.16; 2.63	−5.35 *	−9.22; −1.48
(12 ms vs. t0) Mild	0.38	−26.48; 27.24	−6.41	−14.09; 1.27	−1.62	−4.01; 0.76	−4.23	−12.07; 3.61
(3 ms vs. t0) Moderate	−2.19	−9.09; 4.71	−4.09 *	−7.42; −0.76	−0.45	−1.64; 0.74	−2.41 *	−4.58; −0.23
(6 ms vs. t0) Moderate	0.64	−7.30; 8.58	−2.27	−6.16; 1.63	0.03	−1.33; 1.39	−3.90 **	−6.32; −1.48
(9 ms vs. t0) Moderate	−7.27	−15.61; 1.07	3.54	−7.72; 0.64	−1.16	−2.46; 0.14	−4.19 **	−6.82; −1.57
(12 ms vs. t0) Moderate	−7.02	−16.01; 1.97	−5.33 *	−9.59; −1.07	−1.58 *	−2.90; −0.26	−4.10 *	−6.92; −1.28
(3 ms vs. t0) Severe	−6.26	−20.23; 7.71	−3.05	−8.40; 2.30	−0.85	−2.81; 1.10	−2.79	−6.83; 1.25
(6 ms vs. t0) Severe	−6.25	−21.24; 8.74	−2.11	−8.16; 3.93	−0.78	−2.91; 1.36	−5.50 **	−9.65; −1.35
(9 ms vs. t0) Severe	−15.17 *	−30.57; 0.23	−2.67	−9.71; 4.38	−1.00	−3.40; 1.39	−7.34 **	−11.68;−3.00
(12 ms vs. t0) Severe	−16.74 *	−32.33; −1.15	7.42	−3.11; 17.96	1.26	−2.12; 4.6	−9.36 **	−13.71;−5.01

** *p* < 0.001, * *p* < 0.05. Homa-IR: Homeostatic Model Assessment for Insulin Resistance; HbA1c: Glycosylated Hemoglobin.

**Table 6 nutrients-15-02252-t006:** Generalized Estimating Equation (GEE): expected values for glucose pre- and post-training by time and degree of NAFLD severity.

	Glucose Pre		Glucose Post	
	β	95% CI	β	95% CI
Months of activity				
1	0.00		0.00	
2	1.78	−0.09, 3.65	−2.28 **	−3.62, −0.95
3	−3.20 **	−5.04, −1.36	−2.94 **	−4.28, −1.60
4	1.04	−1.07, 3.15	−2.73 **	−4.23, −1.24
5	−5.81 **	−7.91, −3.71	−2.45 **	−3.99, −0.92
6	−5.76 **	−8.09, −3.44	−3.30 **	−4.99, −1.60
7	−5.30 **	−7.57, −3.02	−5.07 **	−6.70, −3.44
8	−5.23 **	−7.78, −2.68	−4.15 **	−5.98, −2.32
9	−6.13 **	−8.67, −3.59	−2.42 *	−4.27, −0.56
10	−9.36 **	−11.95, −6.78	−3.04 **	−4.94, −1.14
11	−10.28 **	−13.06, −7.50	−5.67 **	−7.69, −3.65
12	−7.33 **	−10.95, −3.70	−0.10	−2.81, 2.61
NAFLD				
Absent	0.00		0.00	
Mild	6.36 **	3.40, 9.32	2.73 *	0.58, 4.88
Moderate	5.18 **	2.49, 7.87	2.26 *	0.29, 4.24
Severe	2.46	−0.78, 5.70	3.60 **	1.21, 0.99

Adjusted for age, sex and BMI. ** *p* < 0.001, * *p* < 0.05. NAFLD: Non-Alcoholic Fatty Liver Disease.

## Data Availability

Data are available from the corresponding author upon request.

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
