# Peer review of "Diet and Exercise Exert a Differential Effect on Glucose Metabolism Markers According to the Degree of NAFLD Severity"

_nutrients, 2023, doi:10.3390/nu15102252_

Round 1
Reviewer 1 Report (Previous Reviewer 2)
It is unfortunate that the authors said they would include extra references, but chose not to. Nonetheless the paper is suitable for publication.
Author Response
"Please see the attachment"

Reviewer 2 Report (New Reviewer)
The study is well-designed, addressed a clinically and epidemiologically important disease with a critical laboratory parameter, HBA1c level in a prevailing disease, NAFLD.
The strength of this therapeutic avenue of combined effect and exercise lies in going in treatment and management beyond, the sole thinking of drug candidates.
Two recommended comments/reviews
1. Methods/Results: In table 3: it should be clearly stated either in legend, methods section, or both that this p value in the last column of the table, in this statistical model, indicates statistical significance among the four categories of NAFLD (including base-line, or the three sub-types (when base-line value is absent) at each each time point of the follow up. i.e. it should be presented clearer to the reader with name of the test, and weather it is vs. base-line or one-way anova.
2. Discussion: One suggestion: for more informative and comprehensive discussion, and since some previous authors reported that exercise and LGIMD Mediterranean diet positively affect the intestinal microbiome (e.g. improve dysbiosis), which in turn improves the glucose metabolism, it would be worth to state this point with one of these citations.
English language is perfect, with sound structure and clear to the reader
Author Response
"Please see the attachment"

Reviewer 3 Report (New Reviewer)
Were there any differences in the patient's family history?
Family history is also important, and if there is a T2DM or NAFLD patient in the family, it may be more difficult to control because there is a high possibility that there will be problems with not only genetic problems but also the family's life style.
How long did it take patients to participate in the current study after they were diagnosed with NAFLD? The severity of symptoms is also important, but if NAFLD has occurred for a long time, it means that there is a reason that is difficult to control, so there is a possibility that it cannot be controlled.
Author Response
"Please see the attachment"

Reviewer 4 Report (New Reviewer)
In the work presented by Antonella Bianco et al., diet and exercise exert a differential effect on Glucose metabolism markers according to the degree of NAFLD severity.
The program of combining diet and exercise introduced by the authors brought good results. The parameter related to glycemia and its levelling has improved in particular (HbA1c). The authors used a sufficient research group (58 subjects). The discussion of the results is exhaustive.
This work has very practical applications. I propose to accept this quality paper and publish it in the Nutrients.
Author Response
"Please see the attachment"

This manuscript is a resubmission of an earlier submission. The following is a list of the peer review reports and author responses from that submission.
Round 1
Reviewer 1 Report
The topic of the present study is of potential interest.
However, I have serious concerns about the methodology of the study.
The effect of an intervention should be evaluated by a randomized controlled trial, not an observational study.
There was no control group with a "traditional" lifestyle approach, therefore the effectiveness of the proposed intervention could not be adequately assessed.
The number of the participants was low and no sample size calculation was performed.
All these conditions heavily affect the validity of the results.
Reviewer 2 Report
Bianco, et. al., detail a small study using exercise and the Mediterranean diet to modulate NAFLD and glucose metabolism. The paper is well written and easy to understand, with a clear hypothesis and appropriate study design. While dietary interventions have been well-studied, the controlled combination of diet and exercise here is exemplary.
However, it would be appropriate to reference more of the previous work done in the area. For example:
Browning, Jeffrey D., et al. "Short-term weight loss and hepatic triglyceride reduction: evidence of a metabolic advantage with dietary carbohydrate restriction." The American journal of clinical nutrition 93.5 (2011): 1048-1052.
The introduction is quite long. I might suggest shortening the paragraph from lines 49-71. I don't think the reader needs to be convinced of the importance of HOMA and HbA1c values in the study of metabolic disease.
Line 196- please choose different wording than "a natural scale." I was immediately left wondering if "log", as in "natural log" had been omitted. The figures are plotted on a linear scale.
Line 343- please make sure to include the group association of the subjects that lost their NAFLD status for clarity in the discussion.
Line 437- This sentence should be amended for clarity. Monitoring HbA1c levels will not prevent the onset of T2D. An intervention if the HbA1c values start to rise could abrogate the onset of T2D.
Reviewer 3 Report
Comments and Suggestions for Authors
In this manuscript, Antonella Bianco et al. recruited 58 subjects with different degrees of NAFLD severity and exposing them to a 12-month period of aerobic exercise combined with dietary treatment, and found that the proposed program significantly improves glucose metabolism parameters, especially HbA1c. Also, the authors stated that monitoring HbA1c levels in non-diabetic subjects may be an important strategy to prevent the onset of T2D in at-risk individuals. Nevertheless, the research conclusions are not enough to be of substantial help to the development of this field. Thus, I don't think this manuscript is suitable for publication in Nutrients.
Here are some specific concerns:
Specific comments
1. Title: The title looks interesting, but it seems to be inconsistent with the study in the text. From the manuscript, it indicates that the authors recruited 58 volunteers and made them follow a 12-month program of combined exercise and diet, the “differential effect” mentioned in the title are not seen in the methods and results, please add some explanation or change the title.
2. Experimental conditions lack standardization: Although, as the authors state, the Mediterranean diet is popular in the region where the experiments were conducted, the current experimental conditions for scientific experiments lack standardization and regulation, and it is difficult to rule out whether individual volunteers adhere to the diet or not.
3. Readability of data: The tables that appear in this manuscript are not very readable, and can easily bring misunderstandings to the reader. For example, in Table 2, the data in parentheses sometimes represent percentages and sometimes represent SD. In addition, does N stand for number? And some unit formats have problems, such as (kg/m2) in table 1 and (MMHG) in table 2. The 95% CI shown in Table 4 appears not to be the normative format either. The object that should be marked with a P value for comparison. Please check and revise the whole manuscript.
4. Data analysis: one-way ANOVA is recommended when multiple group comparisons are made, furthermore, post hoc tests are recommended.
5. The volume of the data is too low: In Table 3, the mild NAFLD group had only one sample size at 12 months, and it is difficult to support the results with such a small sample size.
6. How to explain the clearly increased Homa-IR values in the Severe group in Table 3?